# Expression of Genes Involved in Axon Guidance: How Much Have We Learned?

**DOI:** 10.3390/ijms21103566

**Published:** 2020-05-18

**Authors:** Sung Wook Kim, Kyong-Tai Kim

**Affiliations:** Department of Life Sciences, Pohang University of Science and Technology, Pohang 37673, Korea; kimsw@postech.ac.kr

**Keywords:** axon guidance, local translation, RNA binding proteins

## Abstract

Neuronal axons are guided to their target during the development of the brain. Axon guidance allows the formation of intricate neural circuits that control the function of the brain, and thus the behavior. As the axons travel in the brain to find their target, they encounter various axon guidance cues, which interact with the receptors on the tip of the growth cone to permit growth along different signaling pathways. Although many scientists have performed numerous studies on axon guidance signaling pathways, we still have an incomplete understanding of the axon guidance system. Lately, studies on axon guidance have shifted from studying the signal transduction pathways to studying other molecular features of axon guidance, such as the gene expression. These new studies present evidence for different molecular features that broaden our understanding of axon guidance. Hence, in this review we will introduce recent studies that illustrate different molecular features of axon guidance. In particular, we will review literature that demonstrates how axon guidance cues and receptors regulate local translation of axonal genes and how the expression of guidance cues and receptors are regulated both transcriptionally and post-transcriptionally. Moreover, we will highlight the pathological relevance of axon guidance molecules to specific diseases.

## 1. Introduction

The brain is composed of a complex circuitry of neurons that either send or receive different information acquired from the surrounding environment. These neurons carry out their function through a structure called the synapse, which requires a proper connection between the right axon and the right target [1]. An incorrect or incomplete assembly of the synapse can lead to defects in the neuronal circuit, which can result in different brain disorders like autism [2]. Some of the genome-wide association studies (GWAS) on different neurodevelopmental and neurodegenerative diseases have defined the genes that participate in the guidance and formation of the neuronal circuit as the risk factors [3,4,5,6,7,8]. Hence, proper axon guidance during brain development ensures the formation of a healthy and well-functioning brain.

In the developing brain, there are different molecular cues that guide the axon to its target [9,10]. These molecular cues, which can be categorized as chemo-attractant or -repellent, act simultaneously on the axon to induce various movements, i.e., outgrowth, turning, and collapse [11,12]. The attractants and repellents are known to interact with their receptor partners on the growth cone, the neuron’s motile structure that leads the axon. These attractants and repellents mediate different signaling pathways that modulate the structural dynamics of the growth cone [13,14,15]. Numerous studies have shown that the receptors on the growth cone induce the GTPase-related and PI3K/Akt pathways to alter the dynamics of actin cytoskeleton in response to the extracellular cues [16,17,18]. There are also non-conventional axon guidance cues, such as sonic hedgehog (Shh), bone morphogenetic proteins (BMP), and wingless/int-1 (Wnt) family proteins [19]. These proteins regulate axon guidance through non-canonical signaling pathways. Likewise, a series of events must happen during axon pathfinding to facilitate a precise and accurate interaction between the axon and its target.

The complexity of the neural circuit is far beyond our current understanding, since the neural circuit is formed by billions of neurons that make trillions of synapses [20]. This neural circuit complexity has motivated many scientists to look for more guidance cues, like Draxin, and their receptor partners, hoping to enhance their understanding of neural circuit formation [21]. Other scientists were prompted to study the lesser-known molecular features of axon guidance. Many focused on defining the proteins that participate in the signaling pathway comprising guidance cues and receptors. Some scientists concentrated on the cooperation and crosstalk between different guidance cues while others focused on the expression level of guidance cues and receptors [22,23,24,25,26]. A few scientists illustrated non-canonical pathway of guidance cues and receptors signaling, such as sonic hedgehog pathway or eIF2 pathway [27,28,29,30,31]. In general, extensive research on different molecular aspects of axon guidance, in which the focus of the study is not on the signaling transduction pathway, has also remarkably expanded the current knowledge of neural circuitry development.

For this reason, we will summarize up-to-date findings on the molecular features of axon guidance cues and receptors that extend our knowledge of axon guidance and neural circuit development. In particular, we review recent discoveries on how axon guidance cues and receptors alter the expression of axon guidance-related genes to promote either axon outgrowth or collapse. We will also review how the expression of the guidance cues and receptors are molecularly regulated at different stages of protein synthesis. Moreover, we will highlight the pathological relevance of guidance cues and receptors to diseases like autism spectrum disorder and Alzheimer’s disease.

## 2. Regulation of Local Gene Expression by Guidance Cues and Receptors through RNA Binding Factors

During the late 19^th^ century, a Spanish scientist named Cajal was one of the first scientists to describe the existence of an axon growth cone in the human brain [32], and since then, many scientists have tried to unravel the mystery behind the behavior of axon growth cones in the brain. In the last few decades, scientists have learned that the axon growth cone encounters different axon guidance cues when it travels toward its target [33,34]. Such guidance cues include neurotrophic factors, netrins, semaphorins, ephrins, slit and roundabout. These guidance cues regulate the dynamics of the actin cytoskeleton in the axon growth cone mainly through two signaling pathways, the GTPase-related and PI3K/Akt pathways, which regulate the activity of different downstream proteins [18]. The effects and the mechanisms of the signaling pathways have already been well reviewed [18,34,35]. However, these few signaling pathways seem insufficient to understand the complex system of axon guidance.

Recent studies have shown that there are other molecular aspects to consider in understanding the intricate system of axon guidance. Scientists have demonstrated that the axon guidance cues and receptors also influence the guidance of an axon by regulating the expression of different genes that are related to axon guidance [28,29,36,37]. A recent study showed that the treatment of different guidance cues, like Brain-derived neutrophic factor (BDNF), Netrin-1, or Sema3A, induced a remodeling of the axonal proteome [38]. Furthermore, the treatment of guidance cues up- and down-regulated the levels of different proteins, and these changes were distinct to each guidance cue [38]. These results suggest that the guidance cues and receptors have a variety of roles beyond just inducing different signaling pathways. Hence, for this section, we describe and summarize results from a few recent studies that show how the guidance cues and receptors regulate the expression of different genes that participate in axon guidance (Figure 1).

### 2.1. BDNF Increases the Local Translation of DSCR1.4 mRNA

BDNF is a multi-functioning protein that is essential to both brain development and function. BDNF was initially viewed as a key factor in the survival of sensory neurons [39], but its role in the formation of synapse or in guidance of the axon growth cone was later demonstrated [40]. As an attractive guidance cue, BDNF promotes axon outgrowth through signaling pathways that activate ADF/cofilin, an actin binding protein that regulates the dynamics of actin [41,42]. Recently, BDNF has been found to accelerate the translation of *Down syndrome candidate region 1.4* (*DSCR1.4*) mRNA, and it increases the protein level of DSCR1.4, which is an important neuronal protein that participates in spine morphogenesis and axon outgrowth [29,43,44]. Treatment with BDNF increased the cap-independent translational activity of *DSCR1.4* mRNA in the mouse hippocampal neuron [29]. This regulation was induced by the increased expression of DAP5, an RNA binding protein that promotes cap-independent translation of *DSCR1.4* mRNA [29]. Elevated production of DSCR1.4 stimulated axon outgrowth, while the knock-down of DAP5, and thus the reduction in DSCR1.4, alleviated this effect [29]. This showed that BDNF can coordinate the guidance of an axon by changing the expression of axonal mRNAs through regulating RNA binding proteins (Figure 1b–d).

### 2.2. Netrin-1 Regulates the Local Translation of β-actin, Dscam, and tctp mRNAs

Netrin-1 is another guidance cue that alters the expression of different proteins at the axon growth cone [38]. Netrin-1 was initially known as a chemoattractant, a guidance cue that attracts the axon growth cone [45]. However, later studies revealed that Netrin-1 could function as both a chemoattractant and chemorepellent [46,47,48]. Netrin-1 binds to its receptor, Deleted in Colorectal Cancer (DCC), and induces different signaling pathways that activate either the Rac1 or the TRPC channel [49,50]. Apart from signaling pathways, latest research has shown that Netrin-1 also regulates axon guidance by influencing the local translation of different mRNAs, such as *β-actin, Dscam*, and *tctp* [36,37,51,52,53,54,55].

β-actin is a key cytoskeletal protein that forms the structural component of the axon growth cone. Through the single molecule translation imaging (SMTI) technique, a recent study showed that Netrin-1 increases the de novo protein translation of β-actin [51]. SMTI showed that the translation of Venus-tagged β-actin in the growth cone of retinal ganglion cells was increased by treatment with Netrin-1 [51]. In accordance with this result, other scientists suggested that the increased local translation of β-actin by Netrin-1 is due to an increase in the axonal transport of *β-actin* mRNA toward the growth cone [36]. The transport of mRNA to different subcellular domains, like axon growth cone, can contribute to spatiotemporal specific translation of the mRNA [56]. These results showed that the treatment of Netrin-1 induced the anterograde movement of *β-actin* mRNA while the deletion of the 3′untranslated region (3′UTR) of *β-actin* mRNA abolished this effect [36]. This suggested that the effect of Netrin-1 on the expression of *β-actin* is dependent on its 3′UTR, which infers a possible intervention by other RNA binding proteins [36,57]. In addition, 3′UTR of an mRNA is known to be enriched with *N*^6^-methyladenosine (m^6^A) modification [58]. m^6^A modification can regulate the nuclear export [59] and local translation of mRNAs in axons [60]. In this sense, it would be interesting to find out whether the 3′UTR of *β-actin* mRNA is enriched with m^6^A modification or not. Overall, Netrin-1 clearly regulates the localization and the local translation of *β-actin* mRNA, which is an important factor in axonal structure (Figure 1a).

Down syndrome cell adhesion molecule (DSCAM) is a homophilic cell adhesion molecule that aids in the development of the neuron [61]. It also participates in the connection between the pre- and post-synaptic structure as well as dendritic branching [62,63,64,65]. DSCAM is also known as a Netrin-1 receptor that regulates the turning and extension of the growth cone [66,67,68]. Recently, the local translation of *Dscam* mRNA was found to be regulated by Netrin-1 in the axon growth cone, and the stimulation of hippocampal neurons by Netrin-1 increased the level of DSCAM protein in hippocampal growth cones [37]. DCC, another Netrin-1 receptor, was required for this process. The hippocampal neurons without DCC showed a significantly diminished amount of DSCAM in the axon growth cone [37]. As a result of increased amount of DSCAM, the length of the axon and the number of primary branches were significantly reduced [37]. In general, Netrin-1 increased the local translation of *Dscam* mRNA, which further induced a decrease in axon length and branch number.

Translationally controlled tumor protein (TCTP) is originally known for its role in the promotion of cell growth and survival and the development of cancer [69,70,71]. Recently, the role of TCTP in axon growth and guidance has been elucidated. According to the study, TCTP regulates axon growth in the embryonic visual system, allowing the formation of retinal circuitry [72]. A further study done by the same group revealed that the local translation of *tctp* mRNA was affected by different guidance cues [53]. While ephrin-A1 downregulated the level of TCTP in the growth cone, Netrin-1 upregulated its level. The level of TCTP in the growth cone of retinal ganglion cells was upregulated significantly by the stimulation of Netrin-1. This effect of Netrin-1 was specific to only one of the isoforms of *tctp* mRNA (*tctp-s*), which was defined by a previous study [53], demonstrating that the two isoforms had different 3′UTR sequences [72]. The 3′UTR dependent effect of Netrin-1 on *tctp* local translation indicates the role of other RNA binding proteins whose levels may also be affected by Netrin-1. Overall, Netrin-1 can partake in the axon guidance of retinal ganglion cells by regulating the local translation of *tctp* mRNA.

### 2.3. Sema3A Enhances Local Translation of fmr1 and the Processing of miRNAs 

Semaphorins are the family of secreted and transmembrane proteins that have functions in various parts of our body [73]. Although semaphorins can be organized into eight different classes according to their structure or their origin, semaphorin 3A (Sema3A), a class 3 semaphorin, is the most well studied semaphorin family protein that engages in axon guidance [74]. Sema3A mainly interacts with two families of receptor proteins, Plexin family proteins (Plexin-A1~4) and Neuropilin family proteins (Neuropilin -1 and -2), that are located at the tip of the growth cone and primarily act as a growth cone collapsing cue [75]. As the function of Sema3A has been extensively studied, many axon guidance signaling pathways that are induced by Sema3A are known, such as the RhoA pathway [76] or the ERM family protein pathway [77]. Recently, attention on the function of Sema3A has shifted from its involvement in signaling pathways to its involvement in altering the expression of different proteins. The latest research shows that Sema3A is involved in the local translation of axon guidance-related genes like *fmr1* [27] as well as the processing of many miRNAs that participate in growth cone turning [78].

Fragile X mental retardation protein (FMRP) is an RNA binding protein that regulates the translation of target mRNAs, which control diverse neuronal functions [43,79]. Many of the FMRP target mRNAs are related to either autism or fragile X syndrome (FXS), thus FMRP is also known as a key factor in various neurodevelopmental diseases [80]. FMRP is also involved in axon guidance, as it is required during Sema3A-induced growth cone collapse [81]. The knock-out of *fmr1* in hippocampal neurons completely abolished Sema3A-induced growth cone collapse [81]. Recent results from a study demonstrated that treatment with Sema3A increased the expression of FMRP in the axon growth cone [27]. Sema3A treatment increased the translational activity of *fmr1* mRNA by increasing the protein level of hnRNP Q, a trans-acting factor of internal ribosome entry site (IRES)-mediated translation of *fmr1* mRNA [27]. Also, the knock-down of hnRNP Q in hippocampal neurons decreased the growth cone collapse induced by Sema3A [27]. These results indicated that Sema3A alters the expression of hnRNP Q and thus the level of FMRP that can induce growth cone collapse (Figure 1b–d).

MicroRNAs (miRNA) are small (~22nt) non-coding RNAs that can regulate translation or stability of different transcripts through direct interaction [82]. Some miRNAs are involved in long range axon guidance and axon targeting possibly by regulating axon guidance-related transcripts [83]. For the miRNAs to gain their activity, the miRNAs must be processed from precursor miRNA (pre-miRNA) to mature miRNA. Dicer is an essential protein that produces mature miRNAs from pre-miRNAs [84]. A few studies revealed that both mature miRNA and Dicer were located at the axon growth cone, which inferred growth cone specific maturation of miRNA [85,86,87]. A very recent study demonstrated that this maturation of miRNAs was actually responsive to different guidance cues such as Sema3A or slit2 [78]. The study observed whether treatment with Sema3A induced the maturation of *pre-miR-181a-1* and *pre-miR-181a-2* into *miR-181a-5p, miR-181a-1-3p* and *miR-181a-2-3p*. Treatment with Sema3A on *Xenopus* retinal ganglion cells (RGC) significantly increased the level of mature miRNAs [78]. Through the knock-down system, the study revealed that the aforementioned miRNAs are involved in growth cone collapse. Although it is not clearly detailed yet, Sema3A can indirectly regulate the translation of many axon guidance-related transcripts as it induces the maturation of different miRNAs that regulate mRNA translation and stability. Altogether, not only do different guidance cues and receptors induce signaling pathways, they also alter local translation of numerous axon guidance-related genes to regulate the intricate axon guidance system (Figure 1b–d). 

## 3. Transcriptional and Post-Transcriptional Regulation of the Expression of Guidance Proteins and Receptors

Axon guidance system allows neurons to correctly make numerous synapses with other cells [88,89] so that they can send electrical signals to specific region of the brain [90,91]. Within the axon guidance system, not only are the expression of axonal genes regulated by axon guidance cues and receptors, the expression of guidance cues and receptor themselves are precisely regulated. As the growth cone of an axon travels toward its target, it will encounter different guidance cues, which will bind to receptors on the tip of the growth cone and induce activation of signaling pathways that regulate axon motility [92,93]. During such a process, the expression of guidance cues and receptors must be regulated precisely so that the growth cone of an axon is accurately guided to its target. In this sense, understanding how the expression of guidance cues and the receptors are regulated seems to be especially important for understanding the axon guidance as a whole. Lately, many studies have investigated the regulatory mechanisms behind the expression of guidance cues and receptors [94,95,96,97,98]. Some showed that the expression of guidance proteins and receptors are regulated at different stages of protein synthesis such, as during transcription [99,100]. Thus, for this section, we will introduce and review recent research that illustrates how the expression of different guidance cues and receptors are regulated during both the transcriptional and post-transcriptional processes of protein synthesis (Figure 2).

### 3.1. The Expression of Guidance Cues and Receptors are Regulated by Different Transcription Factors

According to the central dogma of molecular biology, genetic information must go through two important steps during the synthesis of protein. The information contained in DNA is first transcribed into an RNA through the process called transcription. The expression of protein can be regulated during this step through transcription factors, which are the proteins that can either enhance or repress the transcriptional activity of a gene. Transcription of many genes that produce axon guidance molecules are affected by these transcription factors (see Table 1 and Figure 2a).

A recent study showed that transcription factors, like Oct4 and Sox2, and nucleosome remodeling and deacetylase (NuRD) complex repress the transcription of *netrin-1* [95]. Oct4 and Sox2 are well-known transcription factors that are used in the reprogramming of cells [101,102], while the NuRD complex is a well-known epigenetic complex [103]. One study demonstrated that the addition of Oct4 and Sox2 decreased the promoter activity of *netrin-1* [95]. In addition, this reduction in the promoter activity was rescued by the depletion of NuRD complex, which implies the repressive effect of NuRD complex on the transcription of *netrin-1* [95]. In the case of *Drosophila*, dFezf, a *Drosophila* ortholog of the Fezf zinc finger transcription factor [104], regulates the expression of Netrin-A and -B, a *Drosophila* ortholog of Netrin-1 [96,105]. The mRNA levels of *netrin-A* and *–B* were significantly decreased in the dFezf knock-out neurons, which indicates the enhancing effect of dFezf on *netrin-A* and *–B* [96]. Another study demonstrated that the expression of the receptor of Netrin-A and -B, Frazzled/DCC [106], was regulated by Islet [107]. Islet is a well-conserved transcription factor of *Drosophila* that governs axon pathfinding [108]. The study showed that the mRNA level of *fra* in neurons from *isl*-mutated embryos of *Drosophila* was significantly lower than that of the controls [107]. Unlike *Drosophila*, the DCC receptor of the mouse embryo is transcriptionally regulated in an activity-dependent manner [99]. The activator protein 1 (AP-1) binding site [109] was found in the promoter of *DCC,* which is required for the activity-dependent regulation of *DCC* transcription [99]. This also implies that different transcription factors, like the Jun family protein and the Fos family protein, can regulate the transcription of *DCC* [110]. It would be interesting to determine whether this activity-dependent expression of DCC is important in the formation neural circuit. All together, the results from different studies validate the transcriptional regulation of *netrin-1* and its receptor, *DCC,* by different transcription factors.

The expression of semaphorins and their receptor partners are also regulated at the transcriptional level [111]. Sema6A is a semaphorin class-6 guidance protein that regulates angiogenesis [112,113] and the migration of retinal progenitor cells [114]. The transcriptional activity of *sema6a* is regulated by NF-E2-related factor (Nrf2), a transcription factor that has a protective role in response to stress [115]. A recent study revealed that the mRNA level of *sema6a* was significantly increased in Nrf2 knock-out RGC [116]. Also, direct interaction between Nrf2 and *sema6a* was predicted by the analysis of a Nrf2 Chip-seq database [116]. The results inferred the repressive role of Nrf2 on the transcription of *sema6a.* Nrf2 may be a potential therapeutic target for ocular neurodegenerative diseases since regulating the level of Nrf2 may further regulate the Sema6A-mediated migration of retinal progenitor cells [117,118]. Other reports demonstrated transcriptional regulation of few class-3 semaphorin proteins [97,119]. The transcription of *sema3e* is modulated by retinoic acid receptor-related orphan receptor α (RORα), a transcription factor important in circadian rhythm [97,120]. A study illustrated that there were multiple ROR response elements (RORE) on the promoter region of *sema3e*. Direct interaction between RORα and the promoter region of *sema3e* was confirmed through Chip-qPCR analysis. The promoter activity of *sema3e* was negatively affected by the overexpression of RORα, which implied transcriptional repression by RORα [97]. In the case of *sema3a*, SetD5, a transcription factor, was found to regulate the transcription of *sema3a* [119]. SetD5 is a transcription factor that is associated with different neurodevelopmental disorders [121,122,123]. Co-immunoprecipitation (Co-IP) and chromatin immunoprecipitation (ChIP) analyses results from one study showed interaction between SetD5 and BRD2, another transcriptional regulator [124], which further activated transcription of *sema3a* by binding to the *sema3a* promoter region. As such, continued research in transcriptional regulation of many axon guidance genes has revealed the mechanism behind the expression of different guidance molecules, broadening the understanding of axon guidance in general.

### 3.2. RNA Binding Factors Regulate the Translation of Axon Guidance mRNAs

After transcription, a gene’s information is in the form of RNA. A protein is synthesized from translation of the RNA by translational machineries. The expression of proteins can also be regulated during this post-transcriptional stage, through different mechanisms. Many RNA binding proteins in the cell regulate all aspects of RNA, such as its translation, localization, and stability [125,126,127]. These RNA binding proteins bind to different mRNA regions, such as the 5′ untranslated region (5′UTR) and 3′ untranslated region (3′UTR), to either enhance or suppress translational activity. The mRNAs of different guidance cues or receptors are no exception, as some of them were found to be regulated by RNA binding proteins (See Table 1 and Figure 2b) [26,128]. The RNA binding protein Hermes, which is specifically expressed in RGC [129], suppresses the translational activity of *neuropilin-1* mRNA [26]. Neuropilin-1 is a receptor for Sema3A, which induces the growth cone collapse [130]. If the Hermes suppresses the translation of *neuropilin-1* mRNA, the tip of the growth cone would have less Neuropilin-1 for Sema3A to interact with, causing limited growth cone collapse. Also, PUM2, an RNA binding protein that is regulated during synaptic depression [131], increases both stability and translational activity of *neuropilin-1* mRNA by directly binding to the 3′UTR region [128]. As there are about 1400 RNA binding proteins in human [132], the studies on RNA binding protein-mediated translational regulation of other axon guidance cues and receptors seems limitless and may further add meanings to axon guidance system. 

MicroRNA is another factor that is known to regulate the translation or stability of mRNA [133]. As briefly explained previously, miRNAs are small (~22nt) non-coding RNAs that bind to their target mRNAs and further modulate their translation or stability [82]. Recently, few scientists have focused on the role of these miRNAs in the expression of axon guidance cues (Table 1 and Figure 2c) [134,135,136]. *miRNA-92* (*miR-92*), which is a rising biomarker for human cancer [137], regulates the translation of Robo-1, a receptor for the axon guidance cue Slit-1 [18]. *miR-92* represses the translation of *robo-1* mRNA by binding to the miRNA recognition element (MRE) located at the 3′UTR [136]. Other axon guidance cues, like Netrin-1, were found to be translationally regulated by miRNA *let-7* [138]. One study showed that *let-7* directly interacts with the 3′UTR region of *netrin-1* and suppresses its translational activity. The same study also demonstrated that *miR-9* negatively regulates the translation of *DCC* mRNA, a receptor of Netrin-1 [138]. *miR-9* directly binds to the 3′UTR region of *DCC* mRNA and suppresses its translational activity [138]. Although not directly shown, *miR-218* also seemed to repress the expression of DCC, specifically in the dopaminergic neurons of the ventral tegmental area (VTA) [139,140]. The development of dopaminergic neuron in VTA is important in the development of mesocorticolimbic pathway, which is associated with reward-related cognition. The lack of DCC in VTA may delay the maturation of such pathway, causing diseases like schizophrenia and depression [141,142]. Likewise, many other scientists have revealed the regulatory mechanisms behind the translation of different guidance cues and receptor mRNAs, which have widened the knowledge on axon guidance.

## 4. Pathological Relevance

Since axon guidance cues and receptors are critical factors in the formation of neural circuitry, an abnormal expression of or mutation in axon guidance-related genes can lead to several neurological disorders, i.e., neurodevelopmental and neurodegenerative diseases. Many of the axonal genes that are regulated by axon guidance cues and receptors, which were mentioned in previous sections, partake in the formation of synapses. The expression of these genes is critical in maintaining a normal synaptic structure and function. An abnormal expression of these genes may cause a defect in synaptic function and further cause multiple cognitive diseases [143]. Also, the expression of guidance cues and receptors must be regulated precisely during development. An aberrant expression of guidance cues and receptor may cause an incorrect guidance of axon to its target. These erroneous connections between neurons during the development may cause several neurodevelopmental diseases [144]. Many genome wide association studies have identified the association of axon guidance-related genes to different neurological disorders, i.e., autism, Parkinson’s disease, Alzheimer’s disease, and schizophrenia [5,6,8]. A few recent studies have demonstrated that key axon guidance molecules were dysregulated in patients diagnosed with brain disorders [2,145,146]. In this context, we will introduce some of the well-studied axon guidance proteins that are associated with the pathogenesis of autism, and Alzheimer’s disease. 

### 4.1. The Association of Sema5A and Sema3F with Autism 

Autism spectrum disorder (ASD) is a neurodevelopmental disease that mainly causes impairments in social interaction [147]. Although it is difficult to designate genetic mutations as the cause of ASD, the association of defects in different genes with ASD has been extensively studied [148,149,150,151]. For example, *VRK3* is a well-studied gene that is critical for synaptogenesis during neurodevelopment [152]. VRK3 knockout mice showed abnormal expression of TrkB, the BDNF receptor, as well as autistic behaviors [152]. Another study revealed that the deletion of *VRK2* caused abnormal synapse pruning during brain development in mouse, which further exhibited an ASD-like phenotype [153]. Genes that encode axon guidance cues and receptors are also known to be associated with autism [2]. Along with GWAS on autism [154,155], a few recent studies suggested the association of Sema5A and Sema3F with ASD [145,156,157,158]. Sema5A is a class-5 semaphorin guidance molecule that has an important role in neural circuit development [159,160,161], while Sema3F is a class-3 semaphorin guidance molecule that participates in synaptic pruning [162,163]. In the case of Sema5A, a previous study showed that mice lacking Sema5A displayed ASD-like behaviors in social interactions when compared to controls [159]. This led other researchers to investigate if there was any defect in *sema5a* gene of actual patients. A group of scientists found a *de novo* microdeletion and a missense variant of *sema5a* in patients with ASD [4,157], while others found a polymorphism in the *sema5a* promoter region [156]. With Sema3F, the protein is known to interact with numerous ASD-related genes [164,165], so its association with ASD has also been studied. A recent study showed that deletion of *sema3f* could induce ASD-like behaviors in mice [166]. Also, interneuron specific deletion of *sema3f* was enough to induce a deficit in social interaction [145]. Collapsin response mediator protein 2 (CRMP2) is an upstream factor of Sema3F that mediates Sema3F-dependent synaptic pruning [158]. CRMP2 knock-out in mice induced autistic behavior, which suggests that a defect in the regulation of Sema3F signaling can lead to ASD [158].

### 4.2. The Protective Role of Netrin-1 against Alzheimer’s Disease 

Alzheimer’s disease (AD) is the most prevalent form of dementia that is associated with age [167]. It is mainly caused by increased accumulation of toxic amyloid β (Aβ) and neurofibrillary tangles of tau protein in the brain, which leads to the degeneration of synapses and subsequent neuronal cell death [168]. Specifically, Aβ in neurons can cause mitochondrial dysfunction or an increase in the level of reactive oxygen species (ROS) [169,170]. As AD has become a growing threat to many aged individuals, scientists diligently seek a cure for AD; however, no single treatment for AD exists. For over a decade, Netrin-1, an axon guidance cue, has been suggested as a potential treatment for AD due to its role in the production of Aβ [171,172,173]. Netrin-1 can directly interact with amyloid precursor protein (APP) [174] and prevent the production of Aβ [175]. The knock-down of Netrin-1 increased the level of Aβ in AD model mice, while the over-expression of Netrin-1 decreased its level, revealing that Netrin-1 had a protective role against Aβ [175]. It was also recently found that the level of Netrin-1 was decreased in both the serum and cerebrospinal fluid of an Aβ_1-42_-induced AD model rat, which further showed the association of Netrin-1 to the progression of AD [176]. Some scientists further demonstrated that treatment with Netrin-1 improved cognition in AD model mice [171,172]. The injection of Netrin-1 improved the working memory, spatial learning, and spatial memory in AD model mice [171,172]. Although the role of Netrin-1 and its expression in patients with AD have not been studied yet, these studies show that Netrin-1 can potentially be used as the treatment for AD.

## 5. Concluding Remarks

Axon guidance cues and receptors are principal factors in neural circuit development. Furthermore, axon guidance cues and receptors induce different signaling pathways that regulate the dynamics of F-actins, and thus, the motility of the axon growth cone. However, the neural circuit is too complex to be explained by just a few signaling pathways. In this review, we showed that many scientists have tried to expand our current understanding on neural circuit formation by studying different molecular features of the axon guidance system. Many scientists have focused on the regulation of gene expression by axon guidance cues and receptors and its effect on axon guidance. They found that axon guidance cues like BDNF, Netrin-1, and Sema3A changed the local proteome of the axon growth cone by regulating the local translation of axon guidance-participating genes through RNA binding factors. Others have focused on how the expression of these guidance cues and receptors are regulated, revealing that the expression of axon guidance cues and receptors are regulated transcriptionally and post-transcriptionally by transcription factors and RNA binding factors, respectively. Moreover, a few scientists also defined the role of axon guidance cues and receptors in the pathology of brain disorders like autism and Alzheimer’s disease.

Nonetheless, our knowledge on the axon guidance system is still limited and further studies must be done to answer remaining questions. We believe that if we continue to study the different molecular features of axon guidance, we will be a step closer to understanding the axon guidance during development and its related diseases. 

## Figures and Tables

**Figure 1 ijms-21-03566-f001:**
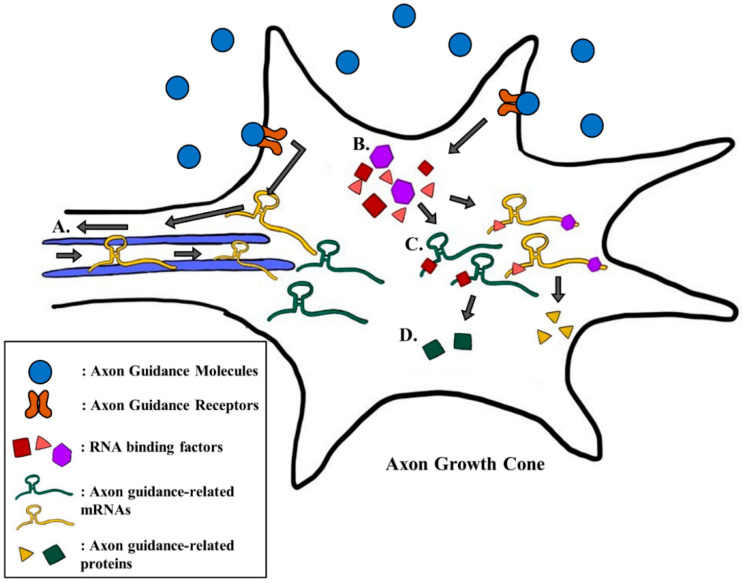
Regulation of local gene expression by axon guidance cues and receptors. The figure summarizes different mechanisms of axon guidance cues and receptor-mediated regulation of local gene expression. (**A**) The axon guidance cues and receptors can induce the localization of specific mRNAs by regulating the 3′UTR-dependent transport system. (**B**) The guidance cues and receptor can also affect the local translation of specific mRNAs by increasing the expression of different RNA binding factors. (**C**) These RNA binding factors, like RNA binding proteins or microRNAs, bind to their target mRNAs and can either enhance or suppress the translational activity of their target mRNAs. (**D**) The level of axonal protein is, therefore, dependent on the interaction between the mRNAs and different RNA binding factors, which is regulated by the axon guidance cues and receptors. The gray arrows represent the pathway of axon guidance-mediated regulation of gene expression.

**Figure 2 ijms-21-03566-f002:**
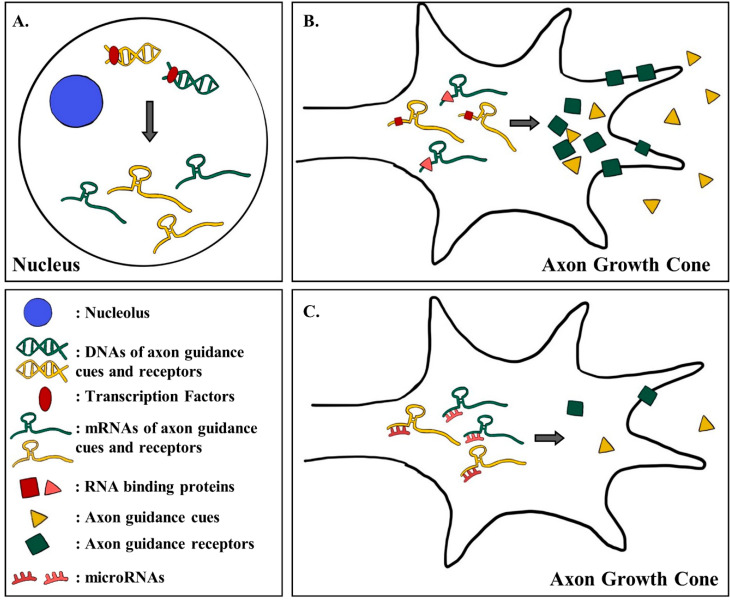
Transcriptional and post-transcriptional regulation of the expression of the guidance proteins and receptors. (**A**) The expression of the guidance cues and receptors are transcriptionally regulated. Different transcription factors bind to the promoter of guidance cues and receptors genes to either enhance or suppress their transcriptional activities. The gray arrow represents the transcription. (**B**) The expression of the guidance cues and receptors are also post-transcriptionally regulated. Different RNA binding proteins bind to the untranslated regions (UTR, 3′UTR and 5′UTR) of the guidance cues and receptors mRNAs to regulate their translational activity or the stability. The gray arrow represents the translation. (**C**) MicroRNAs (miRNAs) can also regulate the expression of guidance cues and receptors post-transcriptionally. Different miRNAs bind to the specific miRNA recognition element (MRE) on the mRNAs of guidance cues and receptors to modulate their translational activity or the stability. The gray arrow represents the translation.

**Table 1 ijms-21-03566-t001:** Summary of factors that regulate gene expression of guidance cues and receptors.

Regulated During:	Targeted Genes	Regulating Factors
Transcription	*netrin-1*	Oct4 [95], Sox2 [95], NuRD [95], dFezf [96]
	*DCC*	Islet [107], AP-1 [99]
	*sema6a*	Nrf2 [116]
	*sema3e*	RORα [97]
	*sema3a*	SetD5 [119]
Post-transcription	*neuropilin-1*	Hermes [26], PUM2 [128]
	*robo-1*	*miR-92* [136]
	*netrin-1*	*let-7* [138]
	*DCC*	*miR-9* [138], *miR-218* [139,140]

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
