# Peer review of "Expression of Genes Involved in Axon Guidance: How Much Have We Learned?"

_ijms, 2020, doi:10.3390/ijms21103566_

Round 1

Reviewer 1 Report

This review covered various areas of axon guidance such as an overview of some of the cues involved, with particular attention to BDNF, netrin and semaphorins, and described the role of local protein synthesis and the involvement of various gene regulatory mechanisms and possible links to disease.

Though generally well written, my main criticism is perhaps the superficial nature of much of the information. I felt that the target audience for this review would be a non-expert audience, looking for general information on the topic, rather than a specialized audience in this field of study. As I don’t know whether this was an invited review or not and/or for what purpose, this may be a moot point.

However, there were many parts of the review where findings were merely listed, without much context, or with very little discussion of the implications of the findings. In other words, I found the work somewhat lacking in perspectives, insights and thought-provoking ideas.

In terms of issues such as lack of context, a few notable examples include the lack of explanation of Sema3a as a predominantly growth cone collapsing cue, and why regulation of DCC in dopaminergic neurons of the VTA might be important (line 309). Furthermore, there are mentions of various proteins and genes (such as Hermes and neuropilin-1; lines 292-296) with very little context of what these are and why they might be important, either in various cell-specific functions, or generally in axon guidance. The authors should ensure not only that the role of various genes and proteins mentioned are put into context, but that certain acronyms (eg. GWAS; line 333) are defined.

The authors also focused on traditional proteinaceous guidance cues, with very little mention of the not-so traditional cues, such as electrical fields, neurotransmitters, retinoic acid etc. Though these might not be the main focus of the review, and there may certainly be less known about their role and/or regulation, they are certainly worth mentioning in the introduction (at least briefly) as other cues known to affect growth cone behaviour and axon guidance.

It was also not made clear why gene regulatory mechanisms and various proteins involved in axon guidance during development, might subsequently be involved in various neurodegenerative disorders later in nervous system functioning. Though an expert in this field would intuitively make this link, the general non-expert audience that this review appears to be aimed at, would not necessarily know this. As such, some statements about this should be included.

There were certain statements and summaries that I found underwhelming and could perhaps have generated further insightful remarks. One example is that netrin’s role in influencing axonal transport of B-actin mRNA requires the 3’UTR. This is hardly surprising information considering the now, quite dated knowledge of the importance of this region for mRNA localization. In fact, there are now some quite interesting studies about how alterations in this region (including its length) might be involved in mRNA localization to axons (vs soma), as well as possible roles for the 5’UTR.  Also – the fact that there is transcriptional regulation of guidance cues and their receptors, is certainly not a ground-breaking revelation.

In summary, I believe this review would be greatly enhanced by providing more context to the literature findings, and certainly more insight and perspectives on the field.

Author Response

We thank this reviewer for allowing us to revise our manuscript and improve the quality of this review. 

Please see the attachment, as we have replied to your comments. 

Thank you.

Reviewer 2 Report

The authors reviewed how local gene expression regulates axon guidance and how a subset of guidance cues could be relevant in neuropathology. The overall structure is slightly peculiar, with a narrative going from circuit formation/axon guidance (section 1), cue-induced local translation (section 2), then back to basic intro on axon guidance again (section 3) and jump back to pre-translational regulation, and then ending with disease relevance. Sections 2 and 3 feel like two independent passages pieced together incoherently and could potentially be improved by switching orders and rewriting.

Throughout the MS, there are many overarching statements without appropriate supports. Also, while citing other reviews are not prohibited, it is not encouraged as it doesn’t provide clarity to the readers and fail to give credits to the original research. Should review papers be cited, they should not be decades old, which could contain inaccurate/outdated models and results that could be misleading to the readers. On the contrary, authors should aim to give credits to original research (even old) that were amongst the first studies to demonstrate results.

Recent literature have convincingly pointed towards that axonal branching, an integral part of axon guidance and circuit formation, is regulated by local translation with both in vitro & in vivo studies, perhaps even more so than axon outgrowth/long-range navigation. However, these branching studies were not mentioned at all in the review. Some key contributions were made by groups such as Gallo, Holt, Schmucker and Twiss.

Overall, the MS can benefit significantly from a more thorough literature review and rewriting.

Minor comments:

  • Line 15-16, what is the difference between signaling pathways and “molecular aspects”?
  • Citation 9: Why cite an outdated 25-year-old review when there are ample high-quality and updated reviews out here, even by the same authors?
  • Citation 20 comes out of the blue and doesn’t match sentence in line 46-48.
  • How true is the statement in line 48-50? “Despite the effort, not many novel guidance cues were found for over a decade. This led other researchers to explore other molecular aspects of axon guidance.”
    • Is there evidence to support this overarching claim?
  • Line 53-53: “A few scientists illustrated non-canonical pathway of guidance cues and receptors signaling”
    • While this statement is true, the authors only cited their own publications without taking into consideration the other important work in the field. Some of which include the Shh pathway and eIF2 pathway, for example.
    • Same for line 78-80.
  • Line 69-70: To support the statement of “in the last few decades”, citations of >25-year-old reviews are again not convincing at all.
  • “RNA binding factors” is not commonly used, do the authors mean RNA binding proteins?
  • For netrin-induced local translation, especially β-actin, it’s important to recognize some important pioneering work by Campbell & Holt, 2001, 2003, Leung et al., 2006.

Author Response

(The authors gave the same response as above.)
